# BOUNDED WORKING MEMORY FOR LLMS: REPRODUCING HUMAN RECALL DYNAMICS

## ABSTRACT

We introduce a cognitively inspired working memory module for large language models (LLMs) that enables efficient narrative recall under capacity constraints. Our approach decomposes input text into structured memory chunks using four methods—semantic, phrase, sentence, and schematic chunking—and integrates prioritization strategies based on salience, connectivity, and temporal decay. These mechanisms enforce a bounded memory capacity, inspired by Miller's number, while preserving information critical for downstream recall. We evaluate the framework on the Naturalistic Free Recall dataset, where models must reconstruct long-form narratives from compressed memory representations. Memory-augmented LLMs achieve higher semantic similarity to human recall transcripts than random baselines, while exhibiting structured retrieval effects such as primacy and recency. These results demonstrate that chunk-based working memory improves the plausibility and efficiency of LLM recall, offering a scalable approach for constrained-context reasoning and memory alignment.

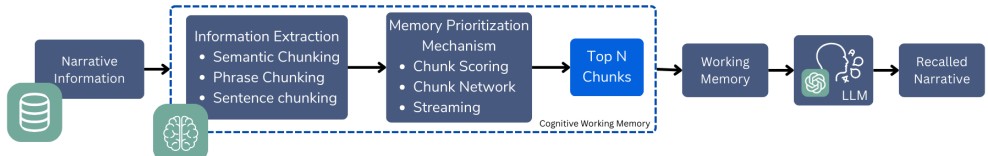

Figure 1: **Cognitive Working Memory.** Narrative input is segmented into chunks and prioritized by salience, connectivity, and recency, ensuring that only cognitively relevant units are retained. The resulting memory buffer forms the basis for reconstructive LLM recall, emulating human working memory dynamics.

## 1 INTRODUCTION

Large language models (LLMs) have achieved remarkable performance across diverse natural language tasks, yet they struggle with a fundamental challenge that humans navigate effortlessly: reasoning and recall under strict memory constraints. While current approaches expand context windows to millions of tokens (Bulatov et al., 2022; Beltagy et al., 2020) or retrieve from vast external databases (Borgeaud et al., 2021), human cognition demonstrates that effective recall emerges not from unlimited storage, but from bounded memory mechanisms that selectively retain, organize, and reconstruct information.

Human working memory has strict capacity limits. A well-known estimate places capacity around $7 \pm 2$ chunks (Miller G, 1956), with later refinements suggesting a lower range of about four chunks depending on context (Cowan, 2001). Despite these limits, humans recall complex narratives through mechanisms such as hierarchical chunking that compresses information into meaningful units (Gobet et al., 2001), event segmentation and schema-driven organization (Baldassano et al., 2017), forgetting mechanisms including interference and replacement (Malleret et al., 2024), and reconstructive recall that infers coherent narratives from sparse traces (Xu et al., 2024). These processes explain why humans exhibit systematic recall patterns, such as primacy and recency effects, even when processing information that far exceeds capacity.

Current LLMs fail to replicate these human-like recall characteristics for two reasons. First, they assume unlimited memory access, leading to inefficient processing where attention becomes diluted as context grows (Chi et al., 2024; Liu et al., 2024). Second, they lack the selective mechanisms that enable humans to prioritize, compress, and reconstruct information under constraints. Existing memory-augmented systems, such as Memory Networks (Weston et al., 2015), Differentiable Neural Computers (Graves et al., 2016), and recent structured approaches like MemTree (Rezazadeh et al., 2025), improve retrieval but do not enforce bounded working memory.

Our central hypothesis is that mimicking human bounded memory can improve recall quality while also providing computational efficiency. To test this, we develop a chunk-based working memory architecture that enforces strict capacity limits through cognitively inspired mechanisms: multi-granular chunking (semantic, syntactic, episodic), salience-based prioritization, temporal decay, and reconstructive generation from compressed memory states.

Our key contributions are threefold:

- We propose a cognitively inspired LLM memory module that enforces bounded working memory capacity using chunking and forgetting mechanisms, in contrast to prior unbounded approaches.
- We demonstrate that our bounded memory system reproduces characteristic human recall behaviors, including primacy and recency effects and reconstructive recall patterns not observed in full-transcript baselines.
- We evaluate on the Naturalistic Free Recall dataset, enabling direct comparison between model outputs and human behavioral data—to our knowledge, the first systematic evaluation of LLM memory systems against human free-recall benchmarks.

Our results show that cognitively constrained memory systems achieve structured, human-like recall while reducing computational load, suggesting that limitations can serve as beneficial inductive biases for generalization in long-context reasoning.

## 2 RELATED WORK

**Human Cognition Mechanisms.** Human working memory operates under strict capacity constraints of about $7 \pm 2$ chunks (Miller G, 1956), refined to approximately four chunks with context-dependent variability (Cowan, 2001). These limits enable efficient processing through mechanisms such as hierarchical chunking that compresses information into meaningful units (Gobet et al., 2001) and event segmentation that parses continuous experience into discrete chunks (Baldassano et al., 2017). Cognitive neuroscience further suggests that working memory capacity is flexibly managed through dynamic gating and interference, where new information actively displaces old through competition (Malleret et al., 2024; Barbosa et al., 2020). Importantly, recall is reconstructive rather than reproductive, shaped by prior knowledge and schemas (Xu et al., 2024; Spens & Burgess, 2024), enabling generalization from sparse memory traces via probabilistic inference processes (Franklin et al., 2020). These findings highlight that bounded memory is not a weakness but a core feature of cognition, allowing efficient, structured recall despite strict capacity limits.

**Memory Models for Long-Context LLMs.** Efforts to extend memory in LLMs generally pursue two directions: unbounded scaling and structured organization, both without bounded working memory. Scaling-based models include the Recurrent Memory Transformer for million-token processing (Bulatov et al., 2022), Longformer with sparse attention (Beltagy et al., 2020), and Transformer-XL with recurrence across segments (Dai et al., 2019). Retrieval-based systems such as RETRO retrieve from trillion-token databases (Borgeaud et al., 2021). State space models like Mamba achieve linear complexity through selective state updates (Gu & Dao, 2024; Dao, 2024), while memory-efficient attention methods such as FlashAttention improve computational throughput (Dao et al., 2022; Dao, 2023).

Structured organization approaches attempt to impose hierarchy or graph structure on stored information. RAPTOR recursively clusters and summarizes content into tree structures (Sarthi et al., 2024), GraphRAG organizes long contexts into graph-based communities (Edge et al., 2025), and MemTree introduces dynamic hierarchical memory that evolves as new information arrives (Rezazadeh et al., 2025). Foundational work such as Memory Networks (Weston et al., 2015) and the Dif-

ferentiable Neural Computer (Graves et al., 2016) established the principle of external memory, but assumed effectively unlimited growth. Cognitive architectures like ACT-R also model competition-based recall but focus on symbolic declarative memory (Stocco et al., 2024).

Despite these advances, current LLMs exhibit fundamental limitations with long contexts: "lost in the middle" degradation (Liu et al., 2024), attention dilution as sequence length increases (Chi et al., 2024), and information over-squashing in deep transformers (Barbero et al., 2024).

From this, we introduce a bounded working memory module for LLMs, the first to be systematically evaluated against human recall data, showing that cognitive constraints enable primacy, recency, and reconstructive recall—establishing memory limits as beneficial inductive biases.

## 3  METHOD

We propose a cognitively inspired chunk-based working memory (WM) module for LLMs that enforces strict capacity constraints. Following Miller's number (Miller G, 1956; Cowan, 2001), WM is modeled as a bounded buffer of size $M$. The module operates in two stages: (i) chunk proposal — extracting candidate units from narrative text using multiple cognitively motivated strategies; and (ii) prioritization and selection — scoring candidates and retaining only the top $M$ for recall. Here we describe the chunk proposal stage.

### 3.1  CHUNKING METHODS

Because the effective unit of human memory is context-dependent and shaped by prior knowledge (Gobet et al., 2001), we implement four complementary chunking strategies. Formally, given a narrative transcript $S = (w_1, \ldots, w_L)$, each chunker defines an extraction function

$$f_{\text{type}} : S \mapsto C^{\text{type}} = \{c_1, \ldots, c_{n_{\text{type}}}\}, \tag{1}$$

where $C_{\text{type}}$ is a set of candidate chunks of that type.

**Semantic Chunking.**  We model semantic clustering (Bousfield, 1953; Collins & Quillian, 1969; Anderson & Bower, 2014) by grouping distributionally similar words. Words are tokenized, stop words removed (NLTK), and embeddings $\{e_i\}$ obtained. For a given seed $w_j$, a semantic chunk is defined as:

$$C_j^{\text{sem}} = \{w_i \mid \cos(e_i, e_j) > \tau, |C_j^{\text{sem}}| \leq k\}, \tag{2}$$

with similarity threshold $\tau = 0.2$ and maximum size $k$. This operationalizes spreading activation models where semantically related items form associative clusters.

**Phrase Chunking.**  Psycholinguistic work suggests humans compress input into phrasal units under temporal constraints (Christiansen & Chater, 2016). Using spaCy dependency parsing, each sentence is analyzed for verb heads $v$ and their arguments (subject $s$, object $o$). Each phrase chunk is an SVO triple:

$$C^{\text{ph}} = (s, v, o), \quad s, v, o \in S. \tag{3}$$

This aligns with the Now-or-Never bottleneck: linguistic input must be rapidly reduced into manageable relational units.

**Sentence Chunking.**  Humans often recall the *gist* of a sentence rather than verbatim content (Baddeley, 2000). We represent this episodic gist as subject–relation–object tuples using the REBEL relation-extraction model:

$$C^{\text{sent}} = (s, r, o). \tag{4}$$

For example, "Barack Obama served as the 44th president of the United States" becomes $C^{\text{sent}} = (\text{Barack Obama}, \text{served as}, \text{president})$. Sentence chunking thus approximates the *episodic buffer* in multicomponent WM theory, which binds information into coherent, retrievable units.

**Schematic Chunking.** Narrative recall is also shaped by schemas (Bartlett & Burt, 1933; Haven, 2007). We segment each story into $N$ episodes and extract structured event elements using a transformer-based event extraction model. Each episode is represented as:

$$E_i = (\text{Character}_i, \text{Goal}_i, \text{Obstacle}_i, \text{Outcome}_i), \quad C^{\text{sch}} = \{E_1, \ldots, E_N\}. \tag{5}$$

These schema-based chunks mirror how humans parse narratives into high-level story frames, supporting efficient encoding and reconstructive recall.

## 3.2 Chunk Prioritization

After candidate chunks are generated, the bounded working memory module must select only $M$ for storage. Selection is guided by three mechanisms: (i) salience-based scoring, (ii) network connectivity, and (iii) streaming replacement dynamics. Together, these mechanisms enforce cognitive plausibility while approximating human recall constraints.

**Salience-Based Chunk Scoring.** Following linguistic salience theories (Boguraev, 1997), we assume that nouns and verbs serve as proxies for informational richness. Each chunk $c_i$ is assigned a salience score:

$$\text{Score}(c_i) = \alpha N_i + \beta V_i + \epsilon_i, \tag{6}$$

where $N_i$ and $V_i$ are the counts of noun and verb tokens within the chunk, $\alpha, \beta \in \mathbb{R}$ are weights, and $\epsilon_i \sim U(0, \delta)$ adds stochasticity. The random noise term simulates variability in human memory and aligns with probabilistic models of cognition and Bayesian retrieval frameworks (Jacobs & Kruschke, 2011).

**Chunk Network Construction.** To capture relationships among chunks, we construct an undirected weighted graph $G = (V, E)$ where each node $c_i \in V$ represents a chunk. Edges encode both semantic similarity and narrative proximity:

$$w_{ij} = \frac{\cos(e_i, e_j)}{1 + |p_i - p_j|}, \tag{7}$$

where $e_i$ is the embedding of chunk $c_i$ and $p_i$ its position in the narrative. This emphasizes semantically similar and narratively close chunks. For each chunk, we compute its strongest connection:

$$W_i = \max_{j \neq i} w_{ij}. \tag{8}$$

Chunks with the top-$k$ $W_i$ values are prioritized for storage, approximating spreading activation in local semantic networks. This mechanism mirrors the CHREST architecture, where chunk formation is guided by familiarity and hierarchical structure (Gobet et al., 2001), and is consistent with neuromorphic and semantic memory models (Li et al., 2016; Jones et al., 2015; Spens & Burgess, 2024).

**Streaming Replacement Mechanism.** To model the temporal dynamics of working memory, we implement a streaming update policy with limited capacity $M = \{m_1, \ldots, m_k\}$, where $k$ is the maximum buffer size. For an incoming chunk $c_t$:

$$\text{if } |M| < k, \quad M \leftarrow M \cup \{c_t\}. \tag{9}$$

Otherwise, a chunk $m_j \in M$ is selected for replacement according to an exponential decay weighting:

$$w_j = \exp(\lambda \cdot j), \quad P_j = \frac{w_j}{\sum_{\ell=1}^{k} w_\ell}, \quad \lambda > 0, \tag{10}$$

where $j = 1$ denotes the oldest and $j = k$ the most recent. A chunk is dropped by sampling from the multinomial distribution $\{P_1, \ldots, P_k\}$, and $c_t$ replaces it. This mechanism retains recent chunks with higher probability, simulating the steep forgetting curve observed in cognitive psychology.

Together, these three mechanisms enforce bounded, selective recall by combining linguistic salience, semantic connectivity, and temporal decay. This provides a cognitively grounded policy for constructing an $M$-slot working memory from an overcomplete set of candidate chunks.

## 3.3 Cognitive Working Memory Model

With chunking strategies and prioritization mechanisms defined, we construct a cognitive working memory (WM) model capable of selecting and storing a bounded set of salient chunks. The model is designed to simulate free recall by making a key cognitive assumption: there is no universally optimal chunking strategy. Instead, individuals exhibit idiosyncratic encoding patterns shaped by personal cognitive processes. To capture this diversity, our framework systematically explores combinations of chunking methods (semantic, phrase, sentence, schematic) with prioritization mechanisms (salience-based, network-based, streaming). Each configuration defines a distinct *memory agent*.

**Participant-Specific Model Fitting.** To evaluate the alignment of memory agents with human recall, we leverage the *Naturalistic Free Recall* dataset, which pairs stories across participants (Pieman–Eyespy and Baseball–Oregon Trail). Since each participant recalls two stories, we adopt a cross-story validation procedure: one story is used to identify the best-fitting agent configuration, and the paired story is used to test generalization. This design avoids data leakage and ensures that models are not simply overfitting to a single narrative.

For example, when modeling a participant who recalled both *Pieman* and *Eyespy*, we first use the *Eyespy* recall transcript as reference. Candidate memory agents are compared by assessing how well the chunks they generate from *Pieman* align with the participant's recall from *Eyespy*. The best-fitting agent is then evaluated on the held-out story, testing whether the chosen chunking–prioritization profile generalizes to novel material.

**Similarity as Cognitive Alignment.** We measure alignment between model-generated chunks and human recall using lexical and embedding-based similarity. Exact word overlap is not required; instead, semantically aligned paraphrases are credited, reflecting reconstructive recall in human memory. This provides a proxy for *cognitive alignment*—the extent to which a bounded-memory agent reproduces recall patterns characteristic of a given participant.

**Reconstructive Generation.** Once a final set of $M$ prioritized chunks is selected, they form the simulated WM buffer. These chunks are then supplied as context to an instruction-tuned LLM with the prompt: *"Reproduce the story based on the given context"*. The LLM's task is to reconstruct a coherent narrative from the compressed representation.

Although the number of available chunks is severely limited, they are selected for semantic richness and narrative salience. This tests whether the LLM can regenerate a full storyline from sparse but informative memory traces, mirroring how humans reconstruct narratives from partial recall. We evaluate the regenerated narratives for both *fidelity* (semantic similarity to the original text) and *generalization* (alignment with human recall patterns).

This integration of chunking, prioritization, and reconstructive generation yields a cognitively inspired memory model that enforces bounded WM constraints while testing the recall capacity of LLMs under psychologically plausible conditions.

## 3.4 Evaluation Metrics

We evaluate the quality of LLM-generated narratives under bounded working memory by combining semantic, event-level, and cognitive-psychology-inspired metrics. These measures assess not only fidelity to the source material but also alignment with human recall patterns from the *Naturalistic Free Recall* dataset.

**Semantic Similarity.** To quantify narrative fidelity, we compute ModernBERTScore, a contextual similarity metric that compares generated narratives against both the original story transcript and the participant's recall. This embedding-based metric captures semantic overlap beyond surface-level token matching, thereby assessing whether the LLM reconstructs meaningful story content from sparse, prioritized chunks. Higher scores indicate stronger alignment with human recall and story semantics.

**Event-Level Recall Probability.** Following the methodology of the *Naturalistic Free Recall* study, each story is segmented into discrete events $E = \{e_1, \ldots, e_T\}$. Generated recall sentences are mapped to events via cosine similarity between sentence and event embeddings. An event is marked as recalled if at least one recall sentence maps to it as the closest match. This yields a binary recall matrix $R \in \{0, 1\}^{N \times T}$, where $R_{i,e} = 1$ if participant $i$ recalled event $e$, and $N$ is the number of participants. Event-level recall probability is defined as:

$$P_{\text{recall}}(e) = \frac{1}{N} \sum_{i=1}^{N} R_{i,e}. \tag{11}$$

We estimate confidence intervals via bootstrap resampling (10,000 iterations), reporting the 2.5th and 97.5th percentiles. To test statistical significance, we perform a permutation test in which participant labels within each column of $R$ are shuffled 10,000 times, yielding a null distribution of recall probabilities. Events whose observed probabilities exceed the 97.5th percentile of this distribution are considered significant.

**Serial Position and Boundary Effects.** To capture hallmark recall phenomena, we compute: (i) the probability of first recall (event first mentioned in the output), (ii) the probability of last recall (final event recalled), and (iii) the serial position curve, which tracks recall likelihood as a function of original event order. Consistent with cognitive psychology (Miller G, 1956; Cowan, 2001), we expect a U-shaped curve reflecting *primacy* (enhanced recall of early events) and *recency* (retention of later events in WM).

**Baseline Comparisons.** To contextualize performance, we evaluated two baselines. In the *random memory assignment* condition, each agent was given randomly sampled chunks matched in length and semantic density but independent of the story or participant, testing whether structured chunking and prioritization improve recall beyond chance. In the *full-transcript recall* condition, agents were provided the complete story transcript without memory constraints, establishing an upper bound on recall performance by simulating unconstrained access to all narrative information.

These baselines together assess whether bounded, cognitively inspired memory mechanisms yield recall behavior that is both more structured than random selection and more human-like than unconstrained full-transcript access.

## 4 RESULTS

### 4.1 MEMORY AGENT SIMULATION AND CHUNK GENERATION

We first evaluate whether memory-augmented LLM agents can mimic human recall patterns using the *Naturalistic Free Recall* dataset, which includes paired participant recalls for *Pieman–Eyespy* and *Baseball–Oregon Trail*. Consistent with the dataset structure, we instantiated 116 memory agents for *Pieman–Eyespy* and 113 for *Baseball–Oregon Trail*. Each agent was paired with the same stimulus as a human participant, with recall simulated through cognitively inspired chunking and prioritization.

Each narrative pair was evaluated bidirectionally: one story served as the target for recall, while its pair served as the reference for constructing the agent's chunking profile—then reversed. This cross-story validation ensured that memory models generalized across narratives rather than overfitting to a single stimulus. Table 1 illustrates representative mappings between dataset participants and their corresponding memory agents, including the working memory configuration and generated chunks.

Following chunk generation, each agent stored between 7 and 9 prioritized chunks, consistent with Miller's number. These memory traces formed the basis for reconstructive recall in the LLM simulation.

### 4.2 LLM RECALL GENERATION AND SEMANTIC FIDELITY

Once the WM buffer was constructed, each agent was prompted to regenerate the target narrative using the stored chunks as context. For all simulations, we used OpenAI GPT-4 with determinis-

| Participant ID | Agent ID | WM Model (Method-Prio-$M$) | Generated Chunks (sample) |
|---|---|---|---|
| P1_pieman | P1 | Schema–Streaming–7 | *Dean McGowan; to replace Fordham's traditionally working- to middle-class students with wealthier, more prestigious ones; Dean McGowan; Dean is covered with cream; Pie Man; Write it up; capeless; fled the scene...* |
| P2_pieman | P2 | Sentence–Scoring–7 | *beer tell me all about it said to be the same as; I gave the figure a name instance of; boo that rule rule instance of; I said, "Yes, Angela." Angela performer...* |

Table 1: **Mapping between dataset participants and memory agents.** Each dataset participant is paired with a memory agent configured using specific chunking and prioritization strategies under bounded working memory ($M = 7$). The table shows example generated chunks illustrating how narratives were segmented into cognitively inspired units.

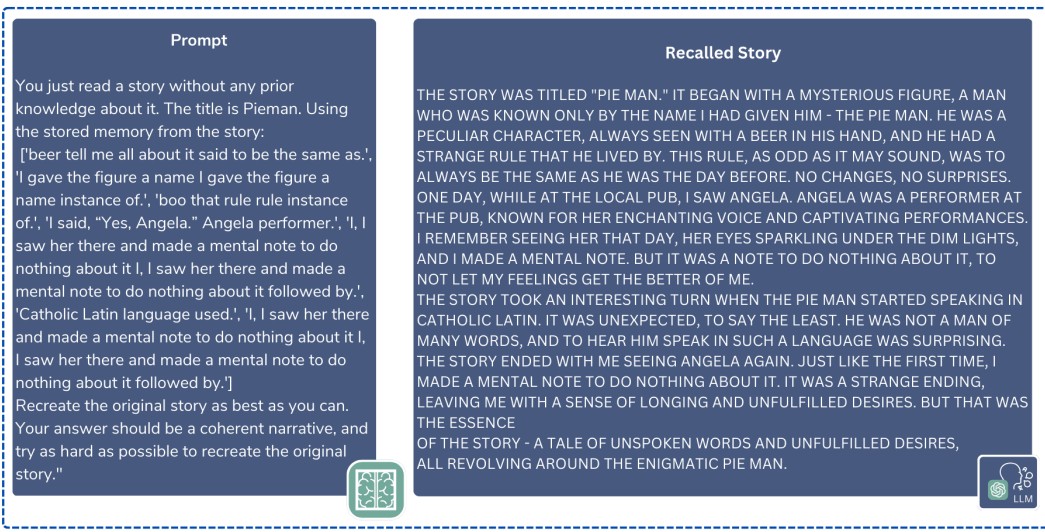

Figure 2: **Prompt with generated chunks and recalled story.** The memory agent receives only 7–9 prioritized chunks. The LLM reconstructs a plausible narrative, filling gaps with schema-driven inference. Errors and interpolations reflect the *Make-Sense Mandate*, where recall emphasizes coherence over verbatim accuracy.

tic decoding (temperature = 0, top_p = 1.0), ensuring reproducibility and attributing performance differences solely to memory model design.

Figure 2 shows an illustrative prompt with generated chunks and the corresponding recalled story. Despite operating with highly compressed and fragmentary memory inputs, the LLM produced coherent narratives that preserved core themes while interpolating missing details. This aligns with the *Make-Sense Mandate* (Haven, 2007), which holds that both humans and machines reconstruct narratives by imposing coherence on incomplete memory traces.

To quantify semantic fidelity, we computed ModernBERTScore between each agent's recall and two references: (i) the original narrative and (ii) the corresponding participant recall. Results are summarized in Table 2. Across all narratives, average similarity between memory-augmented LLM recalls and human transcripts was $0.75$, while similarity to original stories was $0.78$.

Overall, results demonstrate that memory-constrained LLMs produce recalls that are semantically aligned with both human free recall and original texts. While absolute similarity values are moderate due to variability in narrative expression, the closeness of the two distributions indicates that bounded working memory fosters human-like reconstructive recall rather than degrading semantic fidelity.

| Story | WM | Dataset |
|---|---|---|
| Pieman | 0.760 | 0.788 |
| Oregon Trail | 0.728 | 0.738 |
| Eyespy | 0.722 | 0.723 |
| Baseball | 0.767 | 0.784 |

Table 2: **Semantic similarity (ModernBERT).** Alignment of LLM recalls with humans and original stories.

| Story | WM | Human | p-val |
|---|---|---|---|
| Pieman | 37.5% | 37.5% | 1.0000 |
| Oregon Trail | 48.9% | 31.3% | 0.0852 |
| Eyespy | 40.8% | 30.6% | 0.2918 |
| Baseball | 43.6% | 43.6% | 1.0000 |

Table 3: **Significantly recalled events.** Event recall proportions for agents and humans. No significant differences at $\alpha = 0.05$.

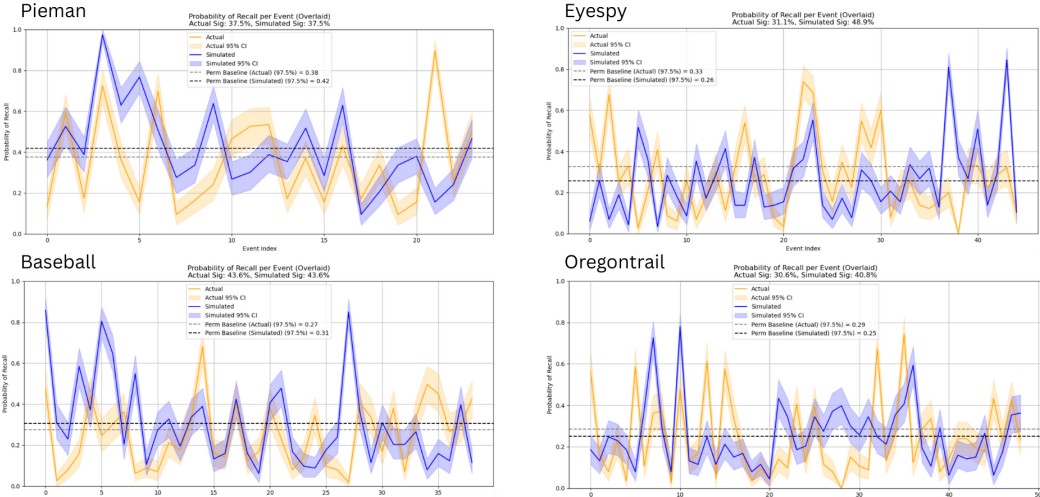

Figure 3: **Event-wise recall probability.** Curves for all four stories with bootstrapped 95% CIs. Dashed lines indicate null baselines from permutation tests.

### 4.3 RECALL PROBABILITY AND TEMPORAL STRUCTURE

We next examined whether memory-augmented LLM agents replicate core recall dynamics observed in human participants. Following the *Naturalistic Free Recall* methodology, we computed event-wise recall probability by mapping generated recalls to predefined story events. A binary recall matrix $R \in \{0, 1\}^{N \times T}$ was constructed, and event recall probabilities $P_{\text{recall}}(e)$ were estimated with 10,000 bootstrap resamples. Statistical significance was assessed using permutation-based null baselines.

Figure 3 shows recall probability curves with 95% confidence intervals. In the *Pieman* story, 37.5% of events exceeded the null baseline, closely matching human recall in the dataset. Across all narratives, proportions of significantly recalled events did not differ from humans at $p < 0.05$ (Table 3), indicating that bounded-memory agents approximate human recall rates without exceeding them.

We further analyzed temporal recall structure. As shown in Figure 4, LLM agents preferentially recalled the first and last events of each story, consistent with primacy and recency effects. Figure 5 reports normalized serial position curves, which exhibit a U-shaped profile characteristic of human recall, with additional mid-story peaks reflecting salient plot points. These findings demonstrate that bounded-memory agents reproduce both event-level selectivity and temporal dynamics of human recall.

### 4.4 BASELINE COMPARISONS

To contextualize performance, we compared structured memory agents with two baselines: (i) random memory assignment and (ii) full-transcript recall (no WM constraint). As expected, the full-transcript condition achieved the highest recall rates (Figure 6). Random memory assignment pro-

Pieman

Eyespy

Baseball

Oregontrail

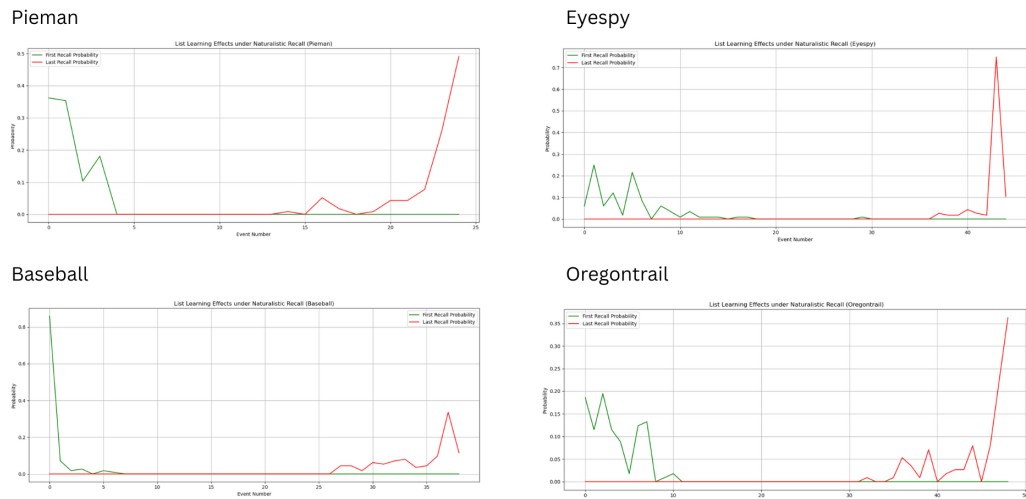

Figure 4: **Primacy and recency effects.** Probability of first and last recall across four stories, showing elevated likelihoods at story boundaries.

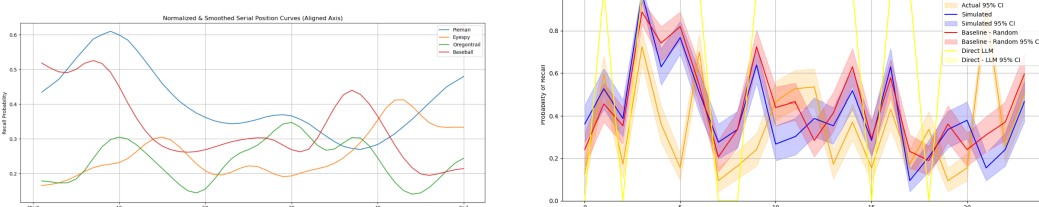

Figure 5: **Serial position curves.** Normalized recall probability across story events. U-shaped profiles capture primacy, recency, and schema-driven mid-story peaks.

Figure 6: **Baseline comparisons.** Recall probability for the *Pieman* story across structured memory, random memory, and full-transcript conditions. Structured memory reproduces human-like selectivity absent in random allocation.

duced recall curves resembling the structured model but lacked consistent event selectivity. In contrast, our bounded-memory agents reproduced hallmark phenomena—primacy, recency, and event salience—absent in random allocation. These results highlight that recall quantity alone is insufficient; cognitive plausibility requires selective, structured memory under capacity constraints.

## 5 CONCLUSION

We presented a cognitively inspired memory module for LLMs that enforces bounded working memory through chunking and prioritization. Our agents reproduced hallmark human recall patterns—including primacy, recency, and reconstructive coherence—while achieving recall performance statistically indistinguishable from humans on the *Naturalistic Free Recall* dataset.

Unlike full-transcript or random baselines, our model demonstrates that cognitive constraints, not recall quantity, drive human-like selectivity. These results establish bounded memory as a psychologically plausible and computationally efficient inductive bias, and they suggest a pathway toward more generalized memory models for LLMs that can support downstream tasks requiring structured, selective recall.

## ETHICS STATEMENT

This work uses the publicly available *Naturalistic Free Recall* dataset, which consists of anonymized human recall transcripts. No personally identifiable information was collected or processed, and the study involves only secondary analysis of existing data. Our experiments focus on modeling memory mechanisms in large language models and do not involve deployment in sensitive or real-world decision-making contexts.

## REPRODUCIBILITY STATEMENT

To ensure reproducibility, we will release all code, model configurations, and evaluation scripts after the blind review process. The release will include detailed instructions to replicate our experiments and extend the proposed framework to related tasks.

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
