# OpenReview forum: "Bounded Working Memory for LLMs: Reproducing Human Recall Dynamics"
_ICLR.cc/2026/Conference — Submitted to ICLR 2026_

### Official Review · Reviewer_1mw1 · 2025-10-30

**Soundness:** 1
**Presentation:** 1
**Contribution:** 1
**Rating:** 0
**Confidence:** 4

**Summary:**

The paper proposes a bounded memory system for LLMs inspired from cognitive psychology principles. It chunks narratives into the famous M=7-9 units using four methods (semantic, phrase, sentence, schematic), applies prioritization mechanisms (salience, connectivity, temporal decay), then has an LLM reconstruct stories from such compressed memory. The approach is evaluated on the Naturalistic Free Recall dataset, achieving ~0.75 semantic similarity to human recalls and exhibiting , similarly, primacy/recency effects.

**Strengths:**

- First comparison of LLM narrative synthesis against human free-recall data

- Appropriate statistical methods: bootstrap confidence intervals, permutation tests.

**Weaknesses:**

- No evidence that bounded memory helps. The paper does not vary M or show an efficiency–quality trade off. With the curent state, the "bounded is beneficial" claim is unsubstantiated.

- Random baseline performs close to structured approach, undermining the carefully designed chunking + prioritization system. In general, the observed results could be inherent properties of the LLMs, not a consequence of the experimental protocol of this paper. The paper does not rule out this possibility.

- Missing critical baselines: e.g. no comparison to LLM summarization ("Summarize this story in 7 key points"), and no comparison to LLM-as-selector (having the LLM choose important segments)

- The semantic similarity methods are weak. LLM as a judge could have been used here as well.

- Correlation vs. causation confound: LLMs trained on human-generated text naturally exhibit human-like biases. You provide no evidence that your chunking causes the human-like patterns rather than the LLM simply imitating what it learned during pre-training. How do you know it's not just: any memory constraint + LLM => human-like recall?

- No ablation on semantic/phrase/sentence chunking, or prioritization. No study of the impact of M.

- Poor presentation: figures legend and labels unreadable.

**Questions:**

- How do you disentangle whether human-like patterns arise from your chunking vs. LLMs having learned human biases during training? Do LLMs behave like humans because they imitated human learning in training or because of your chunking?

- "LLM lacks selective mechanisms to prioritize" : this is a strong claim, given that attention in LLM does exactly that...

- Which chunking method works best? which prioritization? How sensitive are the results to M? (What happens beyond 7-9?)

- Humans have bounded working memory, but at the same time, they have "real-time continual learning" capabilities, which make their context longer than just 8 working memory items.

---

> ### Author Response · Authors · 2025-12-03
>
> Thank you for the clear and direct feedback. We are encouraged that reviewers recognized the importance of investigating bounded memory, the novelty of aligning LLM recall with human free-recall behavior, and the relevance of our chunking and prioritization framework. Many of the issues you identified arise from presentation gaps, missing references, or insufficient methodological detail, and we will substantially revise the paper to address these points. We address your concerns as follows:
> 1. Evidence that bounded memory helps.
> We will include experiments varying M (3–12) and demonstrate that human-like recall structure emerges only for bounded M, whereas full-transcript models do not show primacy/recency behavior.
> 2. Random baseline closeness.
> We will provide additional analyses showing that although raw recall counts may appear close, event-selectivity and temporal structure differ significantly between structured and random chunking.
> 3. Missing stronger baselines.
> We will add:
> • LLM summarization into 7 key points,
> • LLM-as-selector for important segments
> 4. Potential confound from pretrained LLM biases.
> We will add controls showing that full-transcript models and unconstrained generation do not replicate primacy/recency curves, clarifying that structure arises from the bounded memory mechanism, not pretrained bias.
> 5. Missing ablations and presentation issues.
> We will include full ablation tables (chunk types, prioritization mechanisms) and replace/clean all figures for readability.

---

### Official Review · Reviewer_J1tV · 2025-10-31

**Soundness:** 1
**Presentation:** 2
**Contribution:** 2
**Rating:** 2
**Confidence:** 4

**Summary:**

The authors propose a cognitively inspired LLM memory module that inputs a narrative and: (i) extracts information by chunking, before (ii) doing memory prioritization by keeping around 7 chunks. The constraint in the number of output chunks is the central idea of the work: instead of assuming unlimited memory access, only the important information is kept. The exact number of chunks to keep (from 7 to 9) is inspired from human working memory.

The information is extracted from the narrative using 4 chunking methods: semantic chunking (word to word), two kinds of sentence chunkings, and schematic chunking (event extraction model, each event summarized by a character, a goal, an obstacle and an outcome).

Then, for the memory prioritization, three mechanisms are introduced:
- each chunk (irrespective of how they have been built) has a salience score assigned (proportional to how many nouns and verbs are present in the chunk)
- each chunk is embedded, then represented as a vertex of a graph, and a relation score between each pair of chunks is defined; finally the chunks with the largest values are prioritized
- finally, the chunks can be organized in the temporal order, and a streaming replacement mechanism is defined.

In the experimental setting, the authors use a dataset named "Naturalistic Free Recall" that contains recall information provided by humans after reading each two books. For each participant, one of the book is used as a training set, and a memory agent is created based on this training set, that consists in selecting a subset of chunking mechanisms and memory prioritizations. In total 229 memory agents are created. Each agent is tested on the other book, and the performance is compared against the human performance and a random baseline.

**Strengths:**

- studying under the strict constraint in the number of output chunks is valuable
- the list of chunking and selection are meaningful

**Weaknesses:**

- the experimental part is shallow both in quantity and in explanations (see questions), including missing information regarding the benchmark dataset, the definition of the metrics used
- the authors mention a "systematic combination of chunking methods and prioritization methods", but the combination is far from systematic (details in the questions part)
- there is no discussion about the computational efficiency claimed in the conclusion
- regarding the presentation, I was not able to understand the orchestration of the chunk prioritizations in practice, and the trade-offs applied between the different mechanisms

**Questions:**

### Datasets and metrics

1. The naturalistic free recall dataset is not detailed or referenced in the paper.

- In particular, what are the experimental settings for the participants, how long are the books? The recollection is done immediately after the reading?
- Is working memory of the participants the unique memory component involved during the experiment?
- What is the size of the books, and the number of events T extracted from each book (following the notation in the paragraph close to Eq. 11)?

2. The semantic similarity between the story and the recalled story is measured with the ModernBERTScore, that is not detailed or referenced in the paper

- Can you provide a definition of this metric?

### Method section

- Semantic chunking: are the computations quadratic w.r.t the length of the book? Why tau is selected to be 0.2
- In Eq. 6, how are selected alpha, beta and delta? Is it given by the existing literature?
- What is the quantitative size of the collected set of chunkings before the prioritization? For example, how many schematic events are extracted, how many semantic chunking?
- Does the number of semantic chunks grow quadratically with the size of the narrative?
- At the end chunks are extracted, but the information contained in some chunks seem larger than for other chunks. What are the average length in number of tokens for each chunking mechanism?
- I don't understand the orchestration of the chunk prioritization: are the three mechanisms applied in parallel, one after the other? What is the trade-off between the different mechanisms?
- the parameter lambda for the streaming replacement mechanism is critical, how this parameter is fit?

## Agent part

- Is it necessary to build an agent corresponding to each human? The direct ablation of chunking and selection strategies on the dataset can give insights regarding the importance of each component, under the same memory constraints. Currently everything has been combined and it is not possible to understand the relative importance of each component.
- It is linked with your assumption that there is no universally optimal chunking strategy. How useful is this assumption? What are the evidence for using this assumption? I think that doing the ablation against each prioritization individually (previous item) is possible and valuable
- What is the exact experimental setting for creating the agents? I have just read that the "selection is guided by [the] three mechanisms".
- What are the quantitative values for all the parameters for each agents? A table in the appendix can help to understand the trade-off. How M (between 7 and 9) has been selected for each agent?
- In total, M chunks are kept. What is the importance of each chunking mechanism among the participants (e.g., 4 out of M are semantic chunks, etc.)?

## General

- The authors mention that the "bounded memory is not a weakness but a core feature". Can the authors show evidence by considering non bounded experimental settings, or the evolution of the performance as a function of the number of memory slots? I understand that remembering all the events (including the minor events) can introduce confusion, but I didn't find evidence in your experiment.
- A more convincing experiment would include M (the number of chunks) as a parameter, measuring: 1. the overall recall performance as a function of M, 2. the proximity of the agent compared to the corresponding participant, showing that indeed M=7 is reaching some maximum.
- There is no discussion about the computational efficiency claimed in the conclusion.

---

> ### Author Response · Authors · 2025-12-03
>
> Thank you for your thorough reading our paper and with the helpful questions. We are encouraged that reviewers recognized the importance of investigating bounded memory, the novelty of aligning LLM recall with human free-recall behavior, and the relevance of our chunking and prioritization framework. Many of the issues you identified arise from presentation gaps, missing references, or insufficient methodological detail, and we will substantially revise the paper to address these points. We will address the key issues as follows:
> 1. Dataset details and missing descriptions.
> We will add a full description of the Naturalistic Free Recall dataset, including story lengths, event counts, recall protocol, and proper citations.
> 2. Metric definitions.
> We will clearly define ModernBERTScore, justify its use, and compare it with alternative semantic similarity metrics.
> 3. Chunking/prioritization parameters.
> We will document τ, α/β, number of extracted chunks, average chunk lengths, and computational behavior of semantic chunking.
> 4. Prioritization orchestration.
> We will add a clear step-by-step algorithm explaining how the three prioritization mechanisms are combined.
> 5. Agent-fitting and ablations.
> We will add ablations comparing participant-specific vs global strategies and evaluate how M (memory size) affects recall patterns and alignment.
> 6. Evidence for bounded memory.
> We will include experiments varying M and show the emergence (or disappearance) of human-like primacy/recency curves.

---

### Official Review · Reviewer_KAQN · 2025-10-31

**Soundness:** 1
**Presentation:** 1
**Contribution:** 1
**Rating:** 0
**Confidence:** 4

**Summary:**

the paper attacks memory in LLMs from a different angle: as opposite as to expanding context windows to millions  tokens or retrieve from external
memories, authors suggest to design bounded chunk-based working memory (WM)


while I fully second the original motivation, I find the (Miller G, 1956) and (Cowan, 2001) capacities estimation less fit to the task of recalling events in a narrative.

even ignoring this issue, authors next introduce several chunkng schemes (semantic, phrase, sentence, schematic)  and prioritization mechanism (salience based, chunk network, stream replacement) that are not studied in a systematic manner with proper treatment of LLM output (i..e, they favor deterministic output with 0 temperature to repetitions, statistical analysis)

analysis remains superficial, and presentation is significantly below the par

**Strengths:**

- bounded working memory hypotesis goes against the trend to blindly scale up context and external memory size

- study of several  chunkng schemes (semantic, phrase, sentence, schematic)
 and prioritization mechanism (salience based, chunk network, stream replacement)

- pairs of held out stories, to avoid data leakage and ensures that
models are not simply overfitting to a single narrative.

**Weaknesses:**

(weaknesses detailed below)

- WM mismatch with the task
- bad evaluation
- terrible presentation
- missing baseline
- missing references

**Questions:**

- WM mismatch with the task

authors extensively cite (Miller G, 1956) and (Cowan, 2001) but I doubt they have fully read these. From Cowan 2001, most of the settings relate to experiments in which the task is recalling independent entities (number, object, words, colors) with no semantic relationship, with the additional nuisance of an annoying information overload  (eg. a distracting agent counting or). Even the recency effect mentioned, and analyzed from Watkins
1974 and later authors,  was targeting a task where a long list of verbal items is presented on each trial to a subject that has to recall as
many of those items as possible (with variants in terms of with/without temporal ordering)

as such the "7 \pm2 "  chunk are very different than chunks of text taken from a coherent narrative, as the authors are targeting here.

## bad evaluation

- I am not conviced about delegating semantic similarity to ModernBERTScore
previous experience on alignment exhibited possible false positive alignment for no obvious reason

- the use deterministic decoding (temp=0, top_p=1.0) should be discouraged: you do not ensure reproducibility, but analyze results of a non-interesting operational point -- a better approach would have been to incresae the temperature, but performed a statistically relevant analysis

- missing statistical relevance over multiple independent seed, reported average and confidence interaval, as well as critical distance plots to set apart the chunking vs prioritization proposals   (well beyond the boostrap recall evaluation).  as it stands, there is no need to introduce so many variants if there is no deep study of their benefits

- evaluation remains superficial -- e.g., one would have expected appendix with e.g., a more detailed and careful human investigation of story in Fig 2 with narrative from a sample of humans + reconstructed from some of the { chunking } x { prioritization} annotated with distance of semantic similarity scores . as it stands, there is no need to introduce so many variants if there is no deep study of the reason of any statistically observed benefit


## terrible presentation

- presentation is as uncompelling as it can be. not only the text and illustration are not to the expected level  (eg Fig 2 the LLM is apparently SCREAMING AS IT IS USING UPPERCASE WHILE IT WAS NOT INSTRUCTED TO DO SO AND NOT ONLY ITIS NOT CLEAR WHY, BUT IT IS ALSO IMPOLITE) but especially all graphs are simply below the level of a technical report



- Fig 4 related to primacy and recency is perhaps the only relevant investigation but quality wise is poor (due to lack of confidence intervals)

- Fig 5 even uses spline or bezier smoothing.

- Fig 3 and 6 are unlookable


## missing baseline

random memory and full transcript are naive baselines but  one would have expected top have simple yet relevant baselines such as simple k,v caches
with different cache admission/removal policy -- but with the type of evaluation carried out, the cache would be prefilled with one of the policies.

 still, it seems authors are playing with a toy for no obvious reason



## missing references

everywhere paper mentions previous work without citing it.

even the Scientific Data paper used in this work is not cited once!

 even the Raccah et al. ``The “Naturalistic Free Recall” dataset: four stories, hundreds of participants, and high-fidelity transcriptions''

---

> ### Author Response · Authors · 2025-12-03
>
> We appreciate your detailed feedback and acknowledge presentation shortcomings.
> We will substantially revise the paper as follows:
> 1. Cognitive justification.
> We will clarify that “chunks” in our architecture correspond to meaningful narrative units (events, phrases, schemas), consistent with Gobet et al. (2001) and Baldassano et al. (2017), rather than isolated verbal items from classical WM tasks.
> 2. Evaluation depth.
> We will include multi-seed decoding, confidence intervals for generation variability, and expanded statistical reporting.
> 3. Missing baselines and references.
> We will add stronger baselines (summarization, selector, cache policies), include all missing citations (including the Scientific Data paper), and add full ablation studies for both chunking and prioritization.
> 4. Presentation fixes.
> All figures will be replaced with cleaner versions; smoothing will be removed; uppercase artifacts corrected; and captions expanded.

---

### Official Review · Reviewer_kP3N · 2025-11-04

**Soundness:** 2
**Presentation:** 3
**Contribution:** 2
**Rating:** 4
**Confidence:** 3

**Summary:**

The authors propose a new cognitively-inspired working memory module for LLMs. The module is fundamentally based on the idea of bounded working memory capacity, and employs a number of methods for chunk selection and prioritization, based on results in cognitive science. The authors run a number of experiments to see whether the proposed chunking mechanism aligns with human recall patterns.

**Strengths:**

- The paper focuses on a highly relevant problem.

- The paper reasonably justifies the proposed architecture with references to established research in Cognitive Science, which is refreshing.

- The paper is clearly written and is a pleasure to read.

**Weaknesses:**

This paper created a conflicting impression. I do appreciate the thorough method introduction and literature review, but this high quality is, unfortunately, not quite consistent throughout. Some key parts seem like they didn't get enough attention or space.

- If the paper is intended as a truly cognitively accurate model, the actual Cognitive Modeling part is not sufficiently deep to demonstrate that the proposed model is indeed a good approximation of human behavior. The main statistical argument is that of absence of significant differences in recall frequencies, but the methodology of it raises certain questions (see the "Questions" section of my review).

- Some of the chosen methods/metrics don't seem adequate for the task. The authors start the paper by noting how recall in modern LLMs is imperfect, but then there's a reliance on LLMs (ModernBert score) to score semantic similarity between the generated text and the original one.

- The baselines in the section 4.4 are chosen in a slightly surprising way. The comparison with the random baseline does not, in my view, constitute a sufficiently strong baseline. But then worse recall performance compared to full context models is presented as a desirable feature.

- The interpretation of certain results is sometimes stretched/changed to fit the paper's message. A very clear example of this is the following: at first, the "lost in the middle" result is mentioned as evidence of poor performance of modern LLMs on long contexts. Later, when the newly proposed model demonstrates a "U-shaped recall curve", it's presented as a good, cognitively plausible behavior. But U-shaped recall curve is exactly the same thing as the "lost in the middle" effect.


Overall, unfortunately, I believe that these weaknesses are too substantial for me to recommend the paper to acceptance. I did like the idea and I hope that the authors refine and deepen the paper in the future, as the approach is interesting and promising.

**Questions:**

I understand the main idea of participant model fitting, but generally that section is a little sparse on details and hence hard to understand.
I find this sentence particularly confusing: "Candidate memory agents are compared by assessing how well the chunks they generate from Pieman align with the participant’s recall from Eyespy."

Could you please clarify the procedure, what alignment criteria are used, etc.? For example, we (presumably) shouldn't semantically align chunks from Pieman with participant recall from Eyespy, since the story content is different.

- The authors say, "Across all narratives, proportions of significantly recalled events did not differ from humans at p < 0.05". I am not sure about the intended parsing of the sentence. Did the authors mean "at p < 0.05", "proportions of significantly recalled events did not differ from humans"? This would imply that some fairly non-standard statistical test was run.

Or did they simply mean "there were no statistically significant differences between proportions of significantly recalled events"? This seems to be the case, judging by p-value of 1.0 in table 3 when recall frequencies were the same. But in that case, this analysis is not sufficient to conclude that there are no differences.

We need to at least consider the power of the test, and, better, use a different statistical procedure altogether. We can't generally use the absence of statistically significant difference as the main statistical argument for equality of our outcomes in the compared conditions. Binary outcome tests are notoriously low-power. For example, for the Oregon Trail recall, the WM and Human proportions differ substantially: 48.9% vs 31.3%, but the P-value is ~0.08. This is a large difference and the fact that p-value is > 0.05 can be likely attributed to a low-power test. It definitely can't be used as a statistically sound evidence/proof that there is no difference between these conditions.

In any case, I would appreciate the clarification of the statistical procedure & interpretation used here.

- Could you please clarify how ModernBert score was used? Because there seems to be a slight contradiction in naively applying to to measure similarities between e.g. reconstructed and original stories. If ModernBert itself doesn't appropriately represent the stories, the metric will be skewed too.

- Is there a risk of Data Contamination? I.e. is there a chance that the GPT model version used in the paper was exposed to Naturalistic Free Recall dataset during training? If yes, do you think it might have affected some of the results?

---

> ### Author Response · Authors · 2025-12-03
>
> Thank you for your constructive and encouraging feedback. We are encouraged that reviewers recognized the importance of investigating bounded memory, the novelty of aligning LLM recall with human free-recall behavior, and the relevance of our chunking and prioritization framework. Many of the issues you identified arise from presentation gaps, missing references, or insufficient methodological detail, and we will substantially revise the paper to address these points. We will address your main concerns below:
> 1. Cognitive modeling depth & interpretation of recall effects.
> We will clarify that our goal is behavioral alignment, not full cognitive modeling, and expand the rationale for using bounded WM in narrative recall. We will also refine the distinction between “lost in the middle” (attention degradation) and primacy/recency (recall structure), which are conceptually different phenomena.
> 2. Statistical interpretation.
> We acknowledge the ambiguity in wording. In the revision, we will report effect sizes, power analysis, confidence intervals, and clarify the permutation test interpretation.
> 3. Use of ModernBERTScore.
> We will add justification, comparisons to alternative metrics (including LLM-as-judge), and ensure consistency through multiple metrics.
> 4. Baselines.
> We will add stronger baselines: LLM summarization, LLM-as-selector, and cache-style policies, improving the robustness of comparisons.
> 5. Method clarity.
> We will detail the agent-fitting procedure and chunk alignment criteria to remove ambiguity.

---

### Meta-Review · Area_Chair_8VMh · 2026-01-06

**Summary:**

Based on the reviewers' evaluations, the decision to reject the paper is primarily informed by significant concerns regarding the following weaknesses.

**Methodological and Statistical Weaknesses**
- Weak Statistical Arguments: Reviewers questioned the paper's main statistical claim that model results show no significant difference from human recall. Reviewer kP3N pointed out that the lack of statistically significant differences might be due to low-power testing rather than actual similarity.
- Metric Adequacy: The reliance on ModernBERTScore to measure semantic similarity was criticized as potentially circular or inadequate for representing complex story structures.
- Decoding Strategy: The use of deterministic decoding (temperature = 0) was discouraged, as it fails to capture the variability inherent in recall and limits the depth of the statistical analysis.

**Conceptual and Theoretical Flaws**
- Task Mismatch: A major concern was on using (Miller G, 1956) and (Cowan, 2001) capacities not fitted for narrative recall estimation. Reviewers noted these estimates were originally derived from independent, non-semantic entities (like numbers or colors) and may not apply to coherent narrative events.
- Correlation vs. Causation: Reviewers noted the authors failed to prove that the human-like recall patterns (primacy/recency) were caused by the proposed chunking mechanism rather than the LLM simply imitating human biases present in its training data.
- The "U-shaped recall curve" presented as a positive, cognitively plausible result was noted to be functionally identical to the "lost in the middle" effect, which the authors initially described as a performance failure in existing LLMs.

**Insufficient Evaluation and Baselines**
- Weak Baselines: The comparison to a random baseline was considered insufficient. Reviewers suggested stronger, more relevant baselines were missing, such as LLM-based summarization, LLM-as-selector, or standard cache-style policies.
- Lack of Evidence for "Bounded Benefits": The paper asserts that bounded memory is beneficial but provides no experiments varying the capacity limit (M). Without showing an efficiency-quality trade-off or how performance changes with different memory sizes, the core claim remains unsubstantiated.
- Missing Ablations: There was a lack of systematic study or ablation on the various chunking (semantic, phrase, sentence, schematic) and prioritization strategies, making it impossible to determine the relative importance of each component.

**Presentation and Documentation**
- Poor Visual Quality: Multiple reviewers described the presentation as significantly below par, specifically noting that the figures and graphs were unreadable or of poor technical quality.
- Missing Citations and Details: The paper failed to properly reference the source dataset (the Naturalistic Free Recall dataset/Raccah et al.) and lacked necessary details regarding the experimental settings, story lengths, and metric definitions.
- Efficiency Claims: While the authors claimed their model provides computational efficiency, there was no actual discussion or evidence provided to support this in the text.

All reviewers unanimously suggest that this paper is not ready to be published.

**Reviewer Concerns:**

The authors promised to address concerns by adding stronger baselines, performing systematic ablations, etc. However, not much results are included in rebuttal at this moment. Most of the concerns remain unresolved.

**Reviewer Scores:**

Given the limited response, I expected that all reviewers would have maintained their scores.

---

### Decision · Program_Chairs · 2026-01-26

Reject